# Functional characterization of all *CDKN2A* missense variants and comparison to in silico models of pathogenicity

Hirokazu Kimura[1], Kamel Lahouel[2,3], Cristian Tomasetti[2,3], Nicholas Jason Roberts[1,4]*

[1]Department of Pathology, the Johns Hopkins University School of Medicine, Baltimore, United States; [2]Division of Integrated Genomics, Translational Genomics Research Institute, Phoenix, United States; [3]Department of Computational and Quantitative Medicine, Beckman Research Institute, City of Hope, Duarte, United States; [4]Department of Oncology, the Johns Hopkins University School of Medicine, Baltimore, United States

## eLife Assessment

This is a saturation mutagenesis screening of CDKN2A gene, successfully assessing the functionality of the missense variants. The work is **solid** and well-prosecuted. The manuscript was improved during the revision process and this work will serve as a **valuable** resource for diagnostic labs as well as cancer geneticists.

**\*For correspondence:**
nrobert8@jhmi.edu

**Abstract** Interpretation of variants identified during genetic testing is a significant clinical challenge. In this study, we developed a high-throughput CDKN2A functional assay and characterized all possible human *CDKN2A* missense variants. We found that 17.7% of all missense variants were functionally deleterious. We also used our functional classifications to assess the performance of in silico models that predict the effect of variants, including recently reported models based on machine learning. Notably, we found that all in silico models performed similarly when compared to our functional classifications with accuracies of 39.5–85.4%. Furthermore, while we found that functionally deleterious variants were enriched within ankyrin repeats, we did not identify any residues where all missense variants were functionally deleterious. Our functional classifications are a resource to aid the interpretation of *CDKN2A* variants and have important implications for the application of variant interpretation guidelines, particularly the use of in silico models for clinical variant interpretation.

## Introduction

Genetic testing of patients with cancer to identify variants associated with an increased cancer risk and sensitivity to targeted therapies is becoming more common as broad testing criteria are integrated into clinical care guidelines (*Goggins et al., 2020*; *Stoffel et al., 2019*). The American College of Medical Genetics (ACMG) provides a framework to integrate multiple types of evidence, including variant characteristics, disease epidemiology, clinical information, and functional classifications, to interpret variants in any gene (*Richards et al., 2015*). In silico variant effect predictors are also integrated into ACMG variant interpretation guidelines as supporting evidence to aid classification of

variants. While numerous models have been developed, varied accuracy, poor agreement between models, and inflated performance on publicly available data have been reported (*Cubuk et al., 2021*; *Jaffe et al., 2011*; *Wilcox et al., 2022*). Recently developed variant effect predictors aim to overcome these limitations by incorporating deep-learning-based protein structure predictions and by not training on human annotated datasets (*Brandes et al., 2023*; *Cheng et al., 2023*; *Gao et al., 2023*). However, post-development assessment of machine learning-based variant effect predictors, to determine accuracy on novel experimental datasets and suitability for clinical use, are limited.

Variants that cannot be classified as either pathogenic or benign are categorized as variants of uncertain significance (VUSs). However, while pathogenic and benign variants identified during genetic testing are clinically actionable, VUSs are the cause of deep uncertainty for patients and their health care providers as an unknown fraction are functionally deleterious and, therefore, likely pathogenic. For example, individuals with germline VUSs in a pancreatic cancer susceptibility gene are not be eligible for clinical surveillance programs that are associated with improved patient outcomes, unless they otherwise meet family history criteria (*Goggins et al., 2020*; *Stoffel et al., 2019*). Similarly, patients with breast or pancreatic cancer and a germline *BRCA2* VUS would not be eligible for treatment with olaparib, a poly (ADP-ribose) polymerase inhibitor (*Golan et al., 2019*; *Tutt et al., 2021*). Reclassification of VUSs into pathogenic or benign strata has real-world, life-or-death consequences that necessitate a high degree of accuracy.

Germline VUSs in hereditary cancer genes are a common finding in patients with cancer and frequently can be reclassified as pathogenic on the basis of in vitro functional evidence (*Kimura et al., 2022*). In patients with pancreatic ductal adenocarcinoma (PDAC), germline *CDKN2A* VUSs affecting p16$^{INK4a}$, most often rare missense variants, are found in up to 4.3% of patients (*Chaffee et al., 2018*; *Kimura et al., 2021*; *McWilliams et al., 2018*; *Roberts et al., 2016*; *Shindo et al., 2017*; *Zhen et al., 2015*). As functional data from well-validated in vitro assays are incorporated into ACMG variant interpretation guidelines, we recently determined the functional consequence of 29 *CDKN2A* VUSs identified in patients with PDAC using an in vitro cell proliferation assay (*Kimura et al., 2022*; *Richards et al., 2015*). We found that over 40% of VUSs assayed were functionally deleterious and could reclassified as likely pathogenic.

Functional characterization, however, is time-consuming, expensive, and requires technical and scientific expertise. These limitations hinder assessment of in silico variant effect predictors and patient access to functional data that may allow reclassification of VUSs into clinically actionable strata. As *CDKN2A* VUSs will continue to be identified in patients with cancer undergoing genetic testing, we developed a high-throughput functional assay to provide a broad interpretation framework for *CDKN2A* variants. We characterized all possible *CDKN2A* missense variants and compared our functional classifications to recently developed in silico models based on machine learning to determine the accuracy of variant effect predictions.

## Results

### Functional characterization of *CDKN2A* missense variants

We utilized a codon-optimized *CDKN2A* sequence for our multiplexed functional assay. Expression of codon-optimized CDKN2A or the synonymous CDKN2A variants, p.L32L, p.G101G, and p.V126V, in PANC-1, a PDAC cell line with a homozygous deletion of *CDKN2A*, resulted in significant reduction in cell proliferation (p-value<0.0001; *Figure 1—figure supplement 1A*). There was no significant difference between codon-optimized CDKN2A and the three synonymous variants assayed. Conversely, expression of three pathogenic variants, p.L32P, p.G101W, and p.V126D, in PANC-1 cells did not result in any significant changes in cell proliferation. To determine if there were unappreciated selective effects during in vitro culture, we generated a CellTag library based on the pLJM1 plasmid that contained 20 nonfunctional 9 base pair barcodes of equal representation. We then transduced PANC-1 cells that stably expressed codon-optimized CDKN2A with the CellTag library (day 0) and determined representation of each barcode in the cell pool on day 9 and at confluency (day 45). We found no statistically significant changes in barcode representation, indicating that representation of a pool of functionally neutral variants is stable over a period of in vitro culture representing our assay time course (*Figure 1—figure supplement 1B*, *Supplementary file 1*).

We next determined whether we could identify functionally deleterious *CDKN2A* variants at a single residue when all amino acid variants were assayed simultaneously. We generated lentiviral expression plasmid libraries for all 156 CDKN2A amino acid residues, where each library contained all possible amino acids at a single residue. Twenty-seven variants (27 of 3120, 0.87%) were represented in the plasmid libraries at ≤1%. Expression plasmids for each of these 27 variants were individually generated by site-directed mutagenesis and added to the corresponding plasmid library to a calculated representation of 5% (*Figure 1—figure supplement 2A and B*, *Supplementary file 2*). Plasmid libraries were then individually amplified, and lentivirus produced. To confirm that the representation of each variant was maintained after transduction, we transduced three lentiviral libraries (amino acid residues p.R24, p.H66, and p.A127) individually into PANC-1 cells and determined the proportion of each variant in the amplified plasmid library and in the cell pool at day 9 post-transduction. The

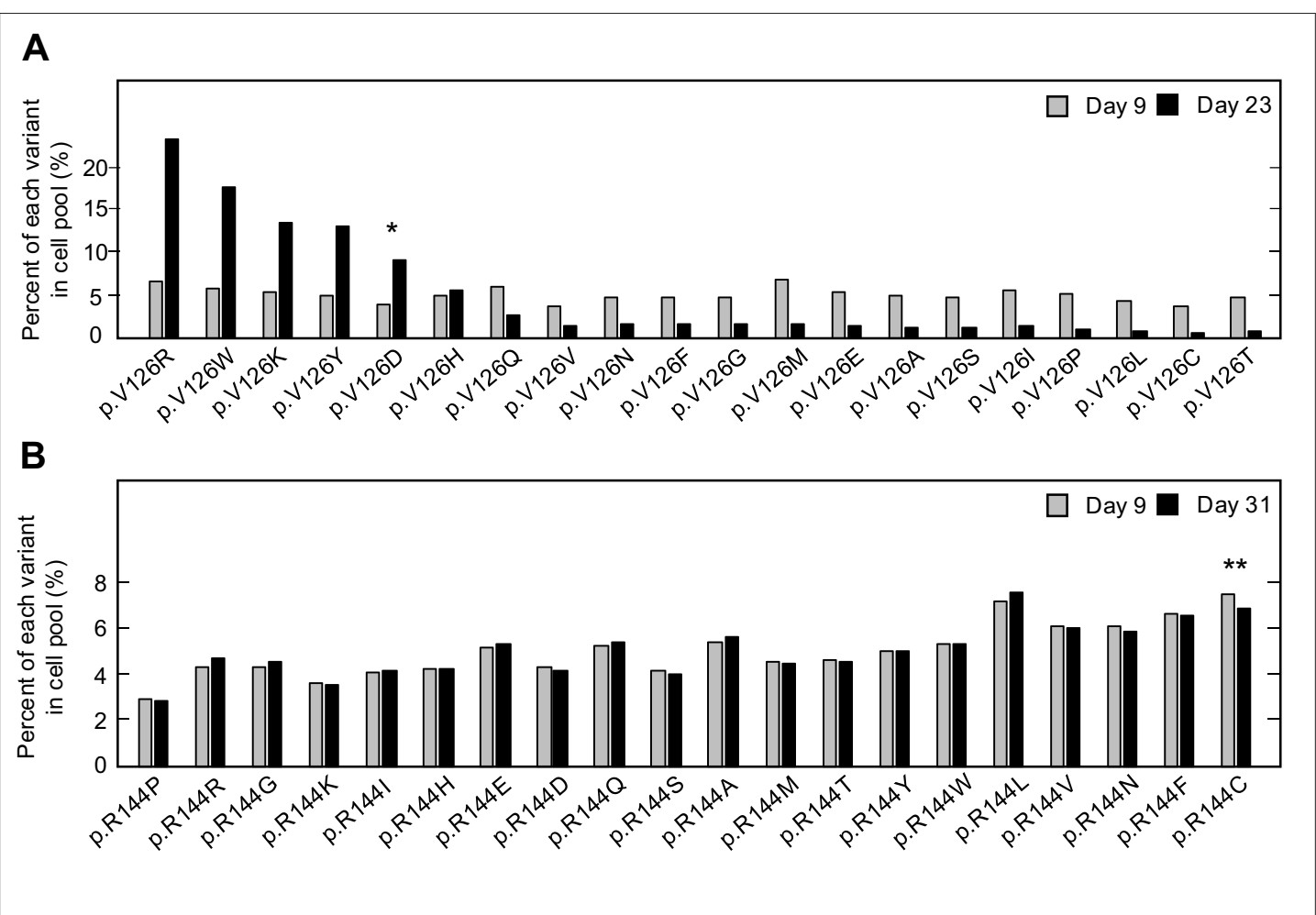

**Figure 1.** Pooled analysis of *CDKN2A* variants at two residues with previously reported pathogenic and benign variants. PANC-1 cell stably expressing 1 of 20 *CDKN2A* variants, 19 missense variants, and 1 synonymous variant, at residue p.V126 or p.R144 were cultured. Variant representation, as the percent of reads supporting the variant sequence, before and after a period in vitro cell proliferation determined by next-generation sequencing for the two residues, p.V126 (**A**) or p.R144 (**B**). *CDKN2A* variant p.V126D (*) was previously reported as pathogenic and increased representation during in vitro proliferation. CDKN2A variant p.R144C (**) was previously reported as benign variant and maintained representation during in vitro proliferation.

The online version of this article includes the following source data and figure supplement(s) for figure 1:

**Source data 1.** Raw data in *Figure 1*.

**Figure supplement 1.** Development and validation of high-throughput CDKN2A functional assay.

**Figure supplement 1—source data 1.** Raw data in *Figure 1—figure supplement 1A*.

**Figure supplement 1—source data 2.** Raw data in *Figure 1—figure supplement 1B*.

**Figure supplement 2.** Data for CDKN2A plasmid library.

proportion of each variant in the amplified plasmid library and in the cell pool at day 9 were highly correlated (*Figure 1—figure supplement 2C and D*, *Supplementary file 3*).

For two CDKN2A amino acid residues that include pathogenic and benign variants, p.V126 and p.R144, we determined the representation of each variant in the transduced cell pool at day 9 and at confluency after a period of in vitro culture, day 23 and day 31 post-transfection, respectively (*Figure 1A and B*, *Supplementary file 4*, *Supplementary file 5*). Two synonymous variants, p.V126V and p.R144R, as well as a previously reported benign variant, p.R144C, either decreased or maintained their representation in the cell pool during in vitro culture as determined by the number of sequence reads supporting the variant. Representation of a previously reported pathogenic variant, p.V126D, increased in the cell pool. Notably, several other variants including p.V126R, p.V126W, p.V126K, and p.V126Y, also increased in representation in the cell pool, suggesting that additional amino acid changes at this residue are functionally deleterious (*Figure 1A*).

To functionally characterize 2964 *CDKN2A* missense variants, PANC-1 cells were transduced with each of the 156 lentiviral expression libraries individually and representation of each *CDKN2A* variant in the resulting cell pool determined at day 9 after transduction and at confluency (days 16–40) (*Supplementary file 5*). Variant read counts were then analyzed using a gamma generalized linear model (GLM), that does not rely on annotation of pathogenic and benign variants to set classification thresholds, and variants with statistically significant p-values were classified as functionally deleterious ($\log_2$ p-values$\leq$–53.2). Variants with p-values that did not reach statistical significance were classified as either of indeterminate function ($\log_2$ p-values>–53.2 and <–5.8) or functionally neutral ($\log_2$ p-values$\geq$–5.8).

We found that 525 of 2964 missense variants (17.7%) were functionally deleterious in our assay (*Figure 2A*, *Figure 2—figure supplement 1A*, *Supplementary file 4*). In addition, 1784 variants (60.2%) were classified as functionally neutral, with the remaining 655 variants (22.1%) classified as indeterminate function (*Figure 2A*, *Supplementary file 4*). In general, our results were consistent with previously reported classifications. Of variants identified in patients with cancer and previously reported to be functionally deleterious in published literature and/or reported in ClinVar as pathogenic or likely pathogenic (benchmark pathogenic variants), 27 of 32 (84.4%) were functionally deleterious in our assay (*Figure 2B*, *Figure 2—figure supplement 1B and C*, *Supplementary file 4*; *Chaffee et al., 2018*; *Chang et al., 2016*; *Horn et al., 2021*; *Hu et al., 2018*; *Kimura et al., 2022*; *McWilliams et al., 2018*; *Roberts et al., 2016*; *Zhen et al., 2015*). Five benchmark pathogenic variants were characterized as indeterminate function, with $\log_2$ p-values from –19.3 to –33.2. Of 156 synonymous variants and six missense variants previously reported to be functionally neutral in published literature and/or reported in ClinVar as benign or likely benign (benchmark benign variants), all were characterized as functionally neutral in our assay (*Figure 2B*, *Figure 2—figure supplement 1B and C*, *Supplementary file 4*; *Kimura et al., 2022*; *McWilliams et al., 2018*; *Roberts et al., 2016*). Of 31 VUSs previously reported to be functionally deleterious, 28 (90.3%) were functionally deleterious and 3 (9.7%) were of indeterminate function in our assay. Similarly, of 18 VUSs previously reported to be functionally neutral, 16 (88.9%) were functionally neutral and 2 (11.1%) were of indeterminate function in our assay (*Figure 2B*, *Figure 2—figure supplement 1B and C*, *Supplementary file 4*).

We next compared variant classifications using the gamma GLM to variant classifications using a normalized fold change method (*Brenan et al., 2016*; *Giacomelli et al., 2018*). Classification of missense variants using normalized fold change also differentiated benchmark pathogenic and benchmark benign variants (*Figure 2—figure supplement 2A and B*, *Supplementary file 6*). Using benchmark pathogenic variants and benchmark benign variants to set thresholds for classification, we classified all variants as either functionally deleterious ($\log_2$ normalized fold change $\leq$0.24), indeterminate function ($\log_2$ normalized fold change >0.24 and <1.09), or functionally neutral ($\log_2$ normalized fold change $\geq$1.09). Using these thresholds, 12 of 18 VUSs (66.7%) previously reported to be functionally neutral were classified as functionally neutral, while 6 (33.3%) were of indeterminate function. Similarly, of 31 VUSs previously reported to be functionally deleterious, 30 (96.8%) were functionally deleterious and 1 (3.2%) was of indeterminate function (*Figure 2—figure supplement 2A and B*, *Supplementary file 6*). Overall, 632 of 2964 missense variants were functionally deleterious (21.3%), 674 variants were indeterminate function (22.7%), and 1658 variants were functionally neutral (55.9%) using $\log_2$ normalized fold change to classify variants (*Figure 1—figure supplement 2C*, *Supplementary file 6*). Notably, 517 of 525 variants (98.5%) classified as functionally deleterious and 1586 of 1784

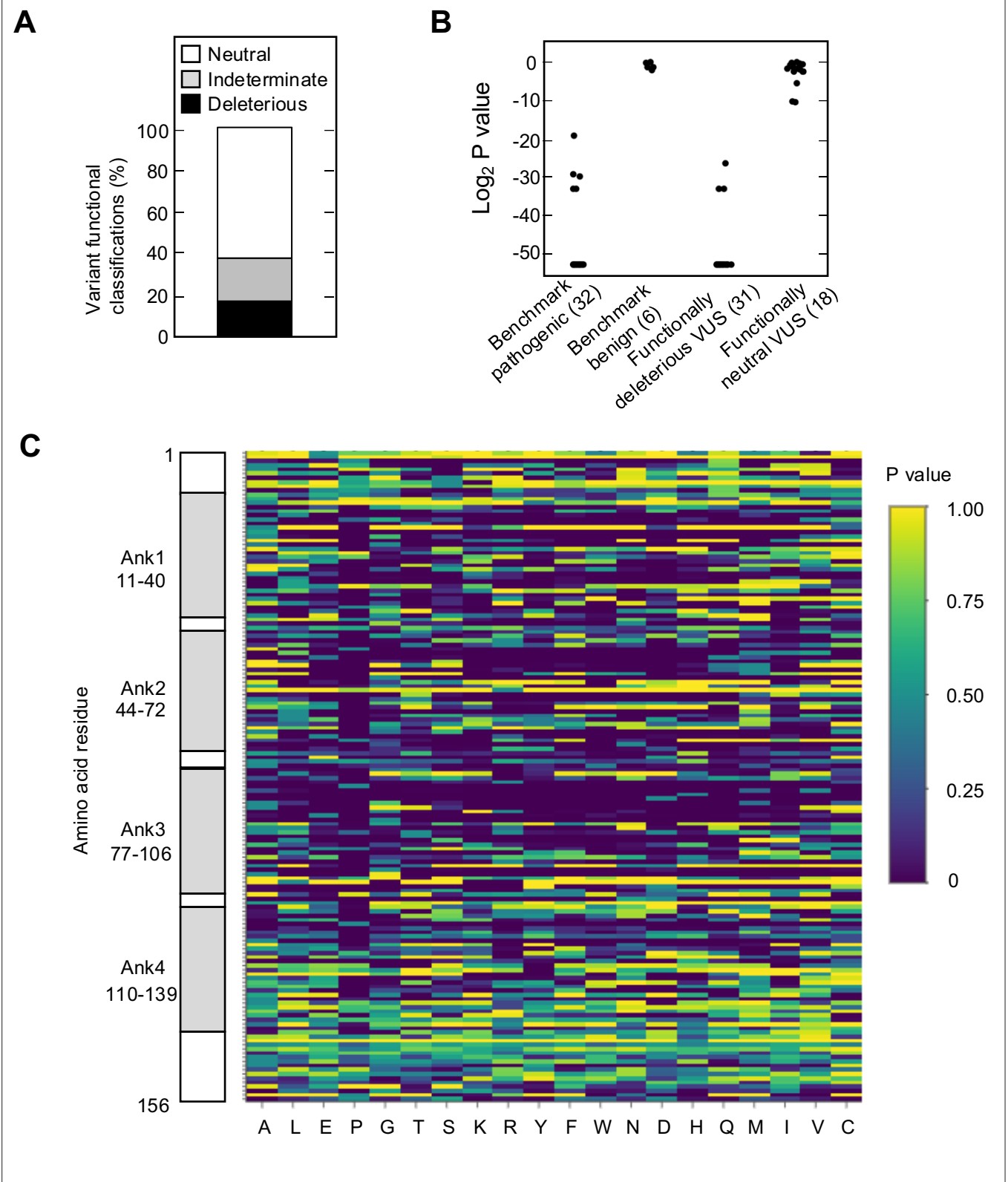

**Figure 2.** Functional characterization of all possible *CDKN2A* missense variants. (**A**) Functional classifications for 3120 *CDKN2A* variants, including 2964 missense variants and 156 synonymous variants. Variants were classified as functionally deleterious, indeterminate function, or neutral based on p-value using gamma generalized linear model (GLM). 525 (17.7%) variants were classified as functionally deleterious. (**B**) Log₂ p-value (gamma GLM) for 32 benchmark pathogenic variants, 6 benign variants, 31 variants of uncertain significance (VUSs) previously reported to have functionally deleterious

*Figure 2 continued on next page*

*Figure 2 continued*

effects, and 18 VUSs previously reported to have functionally neutral effects. (**C**) Heatmap with p-values (gamma GLM) for all 3120 *CDKN2A* variants assayed.

The online version of this article includes the following source data and figure supplement(s) for figure 2:

**Source data 1.** Raw data in *Figure 2B*.

**Figure supplement 1.** p-Values for all possible CDKN2A missense variants.

**Figure supplement 1—source data 1.** Raw data in *Figure 2—figure supplement 1A and B*.

**Figure supplement 2.** Normalized fold change for all possible CDKN2A missense variants.

**Figure supplement 2—source data 1.** Raw data in *Figure 2—figure supplement 2D*.

**Figure supplement 3.** Reproducibility of CDKN2A assay.

**Figure supplement 3—source data 1.** Raw data in *Figure 2—figure supplement 3a*.

**Figure supplement 3—source data 2.** Raw data in *Figure 2—figure supplement 3B*.

**Figure supplement 4.** Proportion of variants in day 9.

**Figure supplement 4—source data 1.** Raw data in *Figure 2—figure supplement 4*.

**Figure supplement 5.** Functional characterization of all possible *CDKN2A* missense variants by ankyrin domain and residue.

**Figure supplement 5—source data 1.** Raw data in *Figure 2—figure supplement 5B*.

**Figure supplement 5—source data 2.** Raw data in *Figure 2—figure supplement 5A*.

variants (88.9%) classified as functionally neutral using the gamma GLM were similarly classified using $\log_2$ normalized fold change (*Figure 2—figure supplement 2D*).

To confirm the reproducibility of our variant classifications, 28 amino acid residues were assayed in duplicate, and variants classified using the gamma GLM. The majority of missense variants, 452 of 560 (80.7%), had the same functional classification in each of the two replicates (*Figure 2—figure supplement 3A and B*, *Supplementary file 4*). Of variants with discordant classifications, 6 (1.1%) were functionally deleterious in one replicate and of indeterminate function in another. While 102 variants (18.2%) were functionally neutral in one replicate and of indeterminate function in another. Importantly, no variant that was functionally deleterious in one replicate and functionally neutral in another (*Supplementary file 4*). Furthermore, the correlation coefficient between duplicate assay results was similar using the gamma GLM and $\log_2$ normalized fold change (*Figure 2—figure supplement 3A and C*). We also determined whether underrepresentation in the cell pool at day 9 affected variant functional classifications. Fifty-three of 2964 missense variants (1.8%) were present in the cell pool at day 9 of the first assay replicate (experiment 1) at <2%, as determined by the number of sequence reads supporting the variant (*Figure 2—figure supplement 4A*, *Supplementary file 4*). There was no statistically significant difference in the proportion of variants classified as functionally deleterious for variants present in less than 2% of the cell pool at day 9 (12 of 53 variants; 22.6%), and variants present in more than 2% of the cell pool (496 of 2911 variants; 17.0%) (p-value=0.28) (*Figure 2—figure supplement 4*). We also found no significant differences in the proportion of variants classified as functionally deleterious for variants present in more than 2% of the cell pool at day 9 when variants were binned in 1% intervals (*Figure 2—figure supplement 4B*).

## Comparison to in silico prediction algorithms

As in silico predictions of variant effect are integrated into ACMG variant interpretation guidelines as supporting evidence, we compared the ability of different algorithms, including recently described algorithms that incorporate deep-learning models of protein structure, to predict the functional consequence of *CDKN2A* missense variants. We compared our functional classifications to predictions from Combined Annotation Dependent Depletion (CADD), Polymorphism Phenotyping v2 (PolyPhen-2), Sorting Intolerant From Tolerant (SIFT), Variant Effect Scoring Tool score (VEST), AlphaMissense, ESM1b, and PrimateAI-3D. In silico predictions for all missense variants were available for PolyPhen-2, SIFT, VEST, AlphaMissense, and ESM1b. For CADD and PrimateAI-3D, 910 (152 functionally deleterious, 196 indeterminate, and 562 functionally neutral) and 904 (152 functionally deleterious, 196 indeterminate, and 556 functionally neutral) missense variants had in silico predictions available respectively (*Supplementary file 7*). In silico variant effect predictors performed similarly across a broad

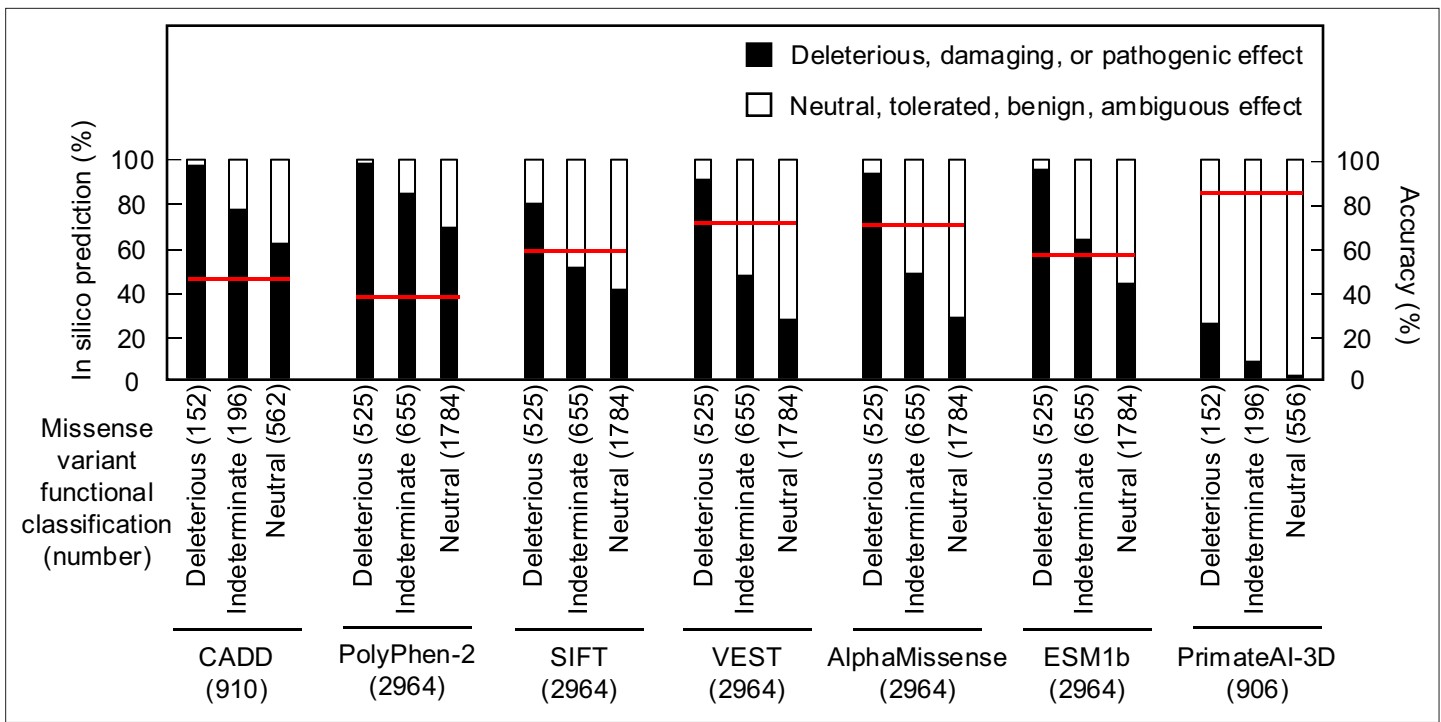

**Figure 3.** Comparison of functional classifications and in silico variant effect predictions for all possible CDKN2A missense variants. Variant effect predictions for *CDKN2A* missense variants using CADD, PolyPhen-2, SIFT, VEST, AlphaMissense, ESM1b, and PrimateAI-3D. Predicted deleterious, damaging, or pathogenic effects (black box) and predicted neutral, tolerated, benign, or ambiguous effects (white box) presented as percent of missense variants with an available prediction. Number of missense variants with an available prediction for each in silico model given in parentheses. Accuracy shown as a red line. CADD: Combined Annotation Dependent Depletion; PolyPhen-2: Polymorphism Phenotyping v2; SIFT: Sorting Intolerant From Tolerant; VEST: Variant Effect Scoring Tool score.

The online version of this article includes the following source data and figure supplement(s) for figure 3:

**Source data 1.** Raw data in *Figure 3*.

**Figure supplement 1.** Variant in silico predictions for seven algorithms.

**Figure supplement 1—source data 1.** Raw data in *Figure 3—figure supplement 1A and B*.

**Figure supplement 1—source data 2.** Raw data in *Figure 3—figure supplement 1C*.

**Figure supplement 1—source data 3.** Raw data in *Figure 3—figure supplement 1D–H*.

**Figure supplement 2.** Variant in silico predictions for five algorithms.

**Figure supplement 2—source data 1.** Raw data in *Figure 3—figure supplement 2A and B*.

**Figure supplement 2—source data 2.** Raw data in *Figure 3—figure supplement 2C*.

**Figure supplement 2—source data 3.** Raw data in *Figure 3—figure supplement 2D and H*.

range of performance characteristics (**Supplementary file 8**). Accuracy of in silico model predictions were 39.5–85.4% (CADD – 45.1%; PolyPhen-2 – 39.5%; SIFT – 60.9%; VEST – 71.9%; AlphaMissense – 71.6%; ESM1b – 59.2%; and PrimateAI-3D – 85.4%) (**Figure 3**). We also assessed sensitivity, specificity, positive predictive value, and negative predictive value for each model. We found that sensitivity was 0.25–0.98 (CADD – 0.97; PolyPhen-2 – 0.98; SIFT – 0.79; VEST – 0.91; AlphaMissense – 0.94; ESM1b – 0.95; and PrimateAI-3D – 0.25), specificity was 0.27–0.98 (CADD – 0.35; PolyPhen-2 – 0.27; SIFT – 0.57; VEST – 0.68; AlphaMissense – 0.67; ESM1b – 0.51; and PrimateAI-3D – 0.98), positive predictive value was 0.22–0.68 (CADD – 0.23; PolyPhen-2 – 0.22; SIFT – 0.28; VEST – 0.38; AlphaMissense – 0.38; ESM1b – 0.3; and PrimateAI-3D – 0.68), and negative predictive value was 0.87–0.98 (CADD – 0.98; PolyPhen-2 – 0.98; SIFT – 0.93; VEST – 0.97; AlphaMissense – 0.98; ESM1b – 0.98; and PrimateAI-3D – 0.87).

We also tested the effect of combining multiple in silico predictors. 904 missense variants had in silico predictions from all seven algorithms. The remaining 2060 missense variants had in silico predictions from five algorithms. Of variants with in silico predictions from all seven algorithms, 378 (41.8%)

had predictions of deleterious or pathogenic effect from a majority of algorithms (≥4), and of these, 137 (36.2%) were functionally deleterious in our assay. Similarly, of 2060 missense variants that had in silico predictions from five algorithms, 1107 (53.7%) had predictions of deleterious or pathogenic effect from a majority of algorithms (≥3), of which, 361 (32.6%) were functionally deleterious in our assay (*Supplementary file 7*).

## Distribution of functionally deleterious variants

Analysis of functionally deleterious variants may highlight critical and non-critical resides for CDKN2A function. We found that functionally deleterious missense variants were not distributed evenly across CDKN2A. CDKN2A contains four ankyrin repeats that mediate protein-protein interactions, ankyrin repeat 1 at codon 11–40, ankyrin repeat 2 at codon 44–72, ankyrin repeat 3 at codon 77–106, and ankyrin repeat 4 at codon 110–139 (*Goldstein, 2004*; *Ruas and Peters, 1998*; *Sun et al., 2010*; *Figure 2—figure supplement 5A*). Functionally deleterious variants were enriched in ankyrin repeat 1 (21.0%, adjusted p-value=0.01), ankyrin repeat 2 (26.2%, adjusted p-value=$1.0 \times 10^{-10}$), and ankyrin repeat 3 (26.3%, adjusted p-value=$2.6 \times 10^{-11}$), while depleted in ankyrin repeat 4 (6.5%, adjusted p-value=$3.2 \times 10^{-13}$) and non-ankyrin repeat regions (6.8%, adjusted p-value=0) (*Figure 2—figure supplement 5B*). Moreover, functionally deleterious variants were further enriched within 10 residue subregions of ankyrin repeats 1–3, with 37.0% of variants in residues 16–25 of ankyrin repeat 1, 40.0% of variants in residues 46–55 of ankyrin repeat 2, and 48.0% of variants in residues 80–89 of ankyrin repeat 3 being classified as functionally deleterious (*Figure 2C*, *Supplementary file 4*).

Across all single residues, the mean percent of functionally deleterious missense variants was 17.7% (95% confidence interval: 12.7–20.9%) (*Figure 2—figure supplement 5C*, *Supplementary file 4*). At five amino acid residues, p.G23, p.G55, p.H83, p.D84, and p.G89, 17 of 19 (89.5%) possible missense variants were functionally deleterious. Notably, these residues are conserved between human and murine p16 (*Byeon et al., 1998*). And p.H83 has been reported to stabilize peptide loops connecting the helix-turn-helix structure of the four ankyrin repeats (*Byeon et al., 1998*), whereas p.D84 and p.G89 are located in a 20-residue region reported to interact with CDK4 and CDK6 (*Fåhraeus et al., 1996*). Conversely, 18 residues were tolerant of amino acid substitutions, with no missense variant characterized as functionally deleterious in our assay (*Figure 2—figure supplement 5C*, *Supplementary file 4*).

We also determined whether the location of variants in protein domains correlated with in silico predictions for the 904 missense variants with predictions from all seven algorithms (*Figure 3—figure supplement 1A–H*) and the 2060 missense variants with predictions from five algorithms (*Figure 3—figure supplement 2A–H*). Notably, Ank2 and Ank3 domains had more variants predicted to have deleterious or pathogenic effect by the majority of algorithms compared to Ank1, Ank4, and non-Ank domains (*Figure 3—figure supplement 1C*, *Figure 3—figure supplement 2C*). We also found increasing agreement between in silico predictions of deleterious or pathogenic effect and functionally deleterious classification in our assay as the number of algorithms predicting deleterious or pathogenic effects increased (*Figure 3—figure supplement 1B*, *Figure 3—figure supplement 2B*). This was true for all CDKN2A protein domains assessed (*Figure 3—figure supplement 1D–H*, *Figure 3—figure supplement 2D–H*).

## Functional effect of *CDKN2A* somatic mutations

Somatic alterations in *CDKN2A* are a frequent finding in many types of cancer. However, not all somatic alterations are unequivocally deleterious to protein function. Missense somatic mutations are particularly challenging to functionally interpret and the presence of a functionally neutral somatic mutation may impact patient care (*Tung et al., 2020*). To understand the functional effect of missense somatic mutations in *CDKN2A*, we functionally classified mutations reported in the Catalogue Of Somatic Mutations In Cancer (COSMIC) (*Forbes et al., 2010*), The Cancer Genome Atlas (TCGA) (*Muddabhaktuni and Koyyala, 2021*), patients with cancer undergoing sequencing at The Johns Hopkins University School of Medicine (JHU), and the Memorial Sloan Kettering-Integrated Mutation Profiling of Actionable Cancer Targets Clinical Sequencing Cohort (MSK-IMPACT) (*Cheng et al., 2015*). Overall, 355 unique missense somatic mutations were reported, of which 119 (33.5%) were functionally deleterious in our assay (*Supplementary file 9*). The percent of missense somatic mutations that were classified as functionally deleterious was greater than the percent of all possible

*CDKN2A* missense variants classified as functionally deleterious, suggesting enrichment of functionally deleterious missense changes among somatic mutations (*Figure 2A*, *Supplementary file 4*, *Supplementary file 9*). The proportion of missense somatic mutations that were functionally deleterious was similar in COSMIC, TCGA, JHU, and MSK-IMPACT. We found that 34.2–53.4% of unique missense somatic mutations classified as functionally deleterious, with 61.4–67.6% of patients having a functionally deleterious somatic mutation (*Figure 4A*, *Supplementary file 9*). As with functionally deleterious variants, functionally deleterious missense somatic mutations were also not distributed evenly across *CDKN2A*, being enriched within the ankyrin repeat 3 (*Figure 4B*, *Supplementary file 9*). We found that 32.4–50.0% of all functionally deleterious missense somatic mutations occurred within ankyrin repeat 3, with 48.0–58.0% of patients in each cohort having a functionally deleterious missense somatic mutation in this domain. Notably, 65.7–76.0% of functionally deleterious missense somatic mutations in this domain were in residues 80–89 (*Supplementary file 9*).

When considering unique missense somatic mutations, 26 of 355 (7.3%) would be classified as pathogenic or likely pathogenic by ACMG classification guidelines and these were found in 263 of 1176 (22.4%) patients in COSMIC, 45 of 185 (24.3%) patients in TCGA, 40 of 184 (21.7%) patients in JHU, and 46 of 174 (26.4%) patients in MSK-IMPACT (*Figure 4—figure supplement 1A and B*). In each cohort, the most prevalent of these somatic mutations were p.His83Tyr and p.Asp84Asn, with more than half of the patients with a somatic mutation that could be classified as pathogenic or likely pathogenic having either the p.His83Tyr or p.Asp84Asn alteration (*Figure 4—figure supplement 1C*). In our functional assays, these somatic mutations were both classified as functionally deleterious.

We were also able to determine the functional classification of *CDKN2A* missense somatic mutations in COSMIC, TCGA, JHU, and MSK-IMAPCT by cancer type. We found that 22.2–100% of *CDKN2A* missense somatic mutations were functionally deleterious depending on cancer type (*Figure 4—figure supplement 2A–D*). When considering missense somatic mutation reported in any database, there was a statistically significant depletion of functionally deleterious mutations in colorectal adenocarcinoma (20.4%; adjusted p-value=$5.4 \times 10^{-9}$) (*Figure 4C*). As the proportion of missense somatic mutations that were functionally deleterious was less in colorectal carcinoma compared to other types of cancer, we assessed whether somatic mutations in mismatch repair genes (*MLH1*, *MLH3*, *MSH2*, *MSH6*, *PMS1*, and *PMS2*) were associated with the functional status of *CDKN2A* missense somatic mutations. Thirty-five patients in COSMIC had a *CDKN2A* missense somatic mutation, of which 12 (34.3%) had a somatic mutation in a mismatch repair gene. We found that no patients with a somatic mutation in a mismatch repair gene had a functionally deleterious *CDKN2A* missense somatic mutation compared to 6 of 23 samples (26.1%) without a somatic mutation in a mismatch repair gene (p-value=0.062).

### *CDKN2A* variants in variant databases

The Genome Aggregation Database (gnomAD) v4.1.0 reports 287 missense variants in *CDKN2A*, including the 13 pathogenic, 4 likely pathogenic, 3 likely benign, 3 benign, and 264 VUSs classified using ACMG variant interpretation guidelines (*Figure 5A and B*, *Supplementary file 10*). Of the 264 missense VUSs, 177 were functionally neutral (67.0%), 56 (21.2%) were indeterminate function, and 31 (11.7%) were functionally deleterious in our assay using the gamma GLM for classification (*Figure 5C*). Similarly, ClinVar reports 395 *CDKN2A* missense VUSs, of which 256 (64.8%) were functionally neutral, 94 (23.8%) were indeterminate function, and 45 (11.4%) were functionally deleterious in our assay (*Figure 5D*, *Supplementary file 11*).

## Discussion

VUSs in hereditary cancer susceptibility genes, predominantly rare missense variants, are a frequent finding in patients undergoing genetic testing and the cause of significant uncertainty. ACMG variant interpretation guidelines incorporate functional data, as well as other evidence such as in silico predictions of variant effect, to aid classification of variants as either pathogenic or benign. *CDKN2A* VUSs are a frequent finding in patients with PDAC. We previously found that over 40% of *CDKN2A* VUSs identified in patients with PDAC were functionally deleterious and therefore could be reclassified as likely pathogenic. In this study, we developed, a high-throughput, in vitro assay and functionally characterized 2964 *CDKN2A* missense variants, representing all possible single amino acid variants. We

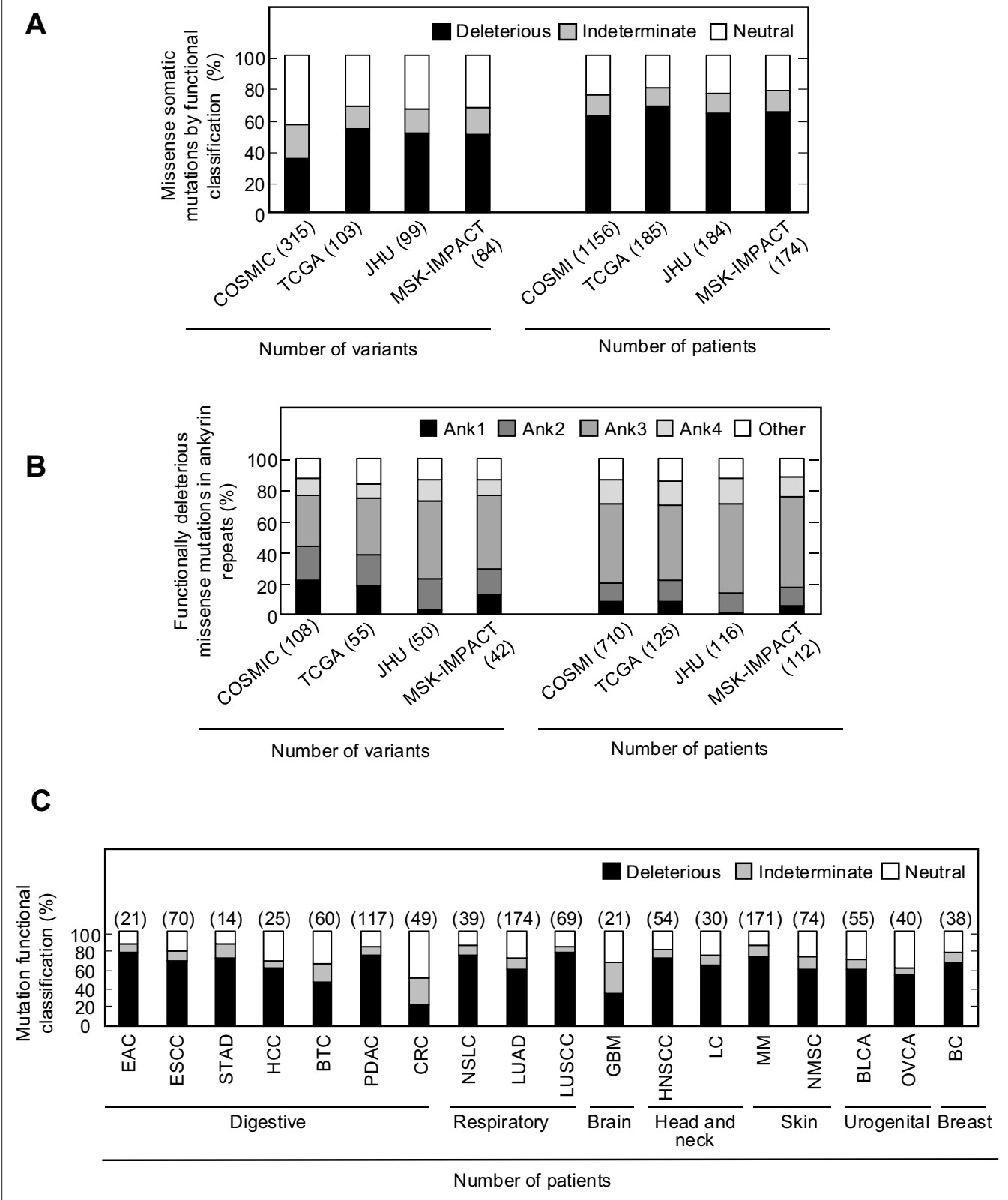

**Figure 4.** Functional classification of missense somatic mutations in *CDKN2A*. (**A**) Somatic missense variants in *CDKN2A* reported in COSMIC, TCGA, JHU, or MSK-IMPACT, by functional classification (deleterious – black box; indeterminate – gray box; neutral – white box). (**B**) Distribution of functionally deleterious missense somatic mutations *CDKN2A* reported in COSMIC, TCGA, JHU, or MSK-IMPACT by ankyrin (ANK) repeat. (**C**) Percent of missense somatic mutations in *CDKN2A* that were classified as functionally deleterious (black box), indeterminate function (gray box), or functionally neutral (white

*Figure 4 continued on next page*

*Figure 4 continued*

box) group by tumor type. Missense somatic mutations reported in COSMIC, TCGA, JHU, and MSK-IMPACT were combined. The number of missense somatic mutations for each tumor type given in parentheses. COSMIC; the Catalogue Of Somatic Mutations In Cancer, TCGA; The Cancer Genome Atlas, JHU; The Johns Hopkins University School of Medicine, MSK-IMPACT; Memorial Sloan Kettering-Integrated Mutation Profiling of Actionable Cancer Targets.

The online version of this article includes the following source data and figure supplement(s) for figure 4:

**Source data 1.** Raw data in *Figure 4A and B*.

**Source data 2.** Raw data in *Figure 4C*.

**Figure supplement 1.** Missense somatic mutations in *CDKN2A*.

**Figure supplement 1—source data 1.** Raw data in *Figure 4—figure supplement 1A and B*.

**Figure supplement 1—source data 2.** Raw data in *Figure 4—figure supplement 1C*.

**Figure supplement 2.** Functional classification of missense somatic mutations in *CDKN2A*.

**Figure supplement 2—source data 1.** Raw data in *Figure 4—figure supplement 2A–D*.

found that 525 missense variants (17.7%) were functionally deleterious. These predefined functional characterizations are resource for the scientific community and can be integrated into variant interpretation schema to aid classification of *CDKN2A* germline variants and somatic mutations.

We classified *CDKN2A* missense variants using a gamma GLM as either functionally deleterious, indeterminate functional, or functionally neutral. However, we did not classify variants that may have gain-of-function effects, resulting in decreased representation in the cell pool. Future studies are necessary to determine the prevalence and significance of *CDKN2A* gain-of-function variants.

Importantly, variant classifications using a gamma GLM were not biased by assay outputs for previously reported – benchmark – pathogenic or begin variants to determine thresholds. Classification thresholds were determined using the change in representation of 20 nonfunctional barcodes in a pool of PANC-1 cells stably expressing CDKN2A after a period of in vitro proliferation. Even so, *CDKN2A* missense variant classifications were remarkably similar using a gamma GLM or normalized fold change with thresholds determined using benchmark pathogenic and begin variants. Of missense variants classified as functionally deleterious using a gamma GLM, 98.5% were similarly classified using normalized fold change.

We repeated our functional assay twice for 28 CDKN2A residues. For the remaining 128 residues of CDKN2A, the functional assay was completed once. While we found general agreement between functional classifications from each replicate for the 28 residues assayed in duplicate, additional repeats for each residue are necessary to determine variability in variant functional classifications.

Our characterization of all possible *CDKN2A* missense variants allowed us to assess the ability of in silico algorithms – including recently published predictors based on machine learning AlphaMissense, ESM1b, and PrimateAI-3D – to predict the pathogenicity or functional effect of *CDKN2A* missense variants. We found that all in silico variant effect predictors assessed performed similarly. Highest accuracy was observed with PrimateAI-3D at 85.4%, followed by VEST at 71.9% and AlphaMissense at 71.6%. Importantly, even in silico predictors performing best in one metric may perform poorly in others. For example, PrimateAI-3D had the highest specificity (0.98) and positive predictive values (0.68), but the lowest sensitivity (0.25) and negative predictive value (0.87). Given that reclassification of VUSs in hereditary cancer genes into inappropriate strata has significant implications for patients, use of in silico models for clinical variant interpretation, including those utilizing machine learning, may be premature. Ultimately, our data support current ACMG guidelines that include in silico predictions of variant effect as supporting evidence of pathogenicity or benign impact.

Our study also provides other insights for the implementation of variant interpretation guidelines. ACMG guidelines include presence of a missense variant at a residue with a previously reported pathogenic variant as moderate evidence of pathogenicity. We found that functionally deleterious missense variants were not evenly distributed across *CDKN2A*. We found enrichment of functionally deleterious missense variants in ankyrin repeats 1–3 and depletion in ankyrin repeat 4. Notably, no CDKN2A residue was completely intolerant of amino acid changes. Suggesting, at least for *CDKN2A*, that the presence of a pathogenic missense variant at a residue should be used with caution when classifying other missense variants at the same residue.

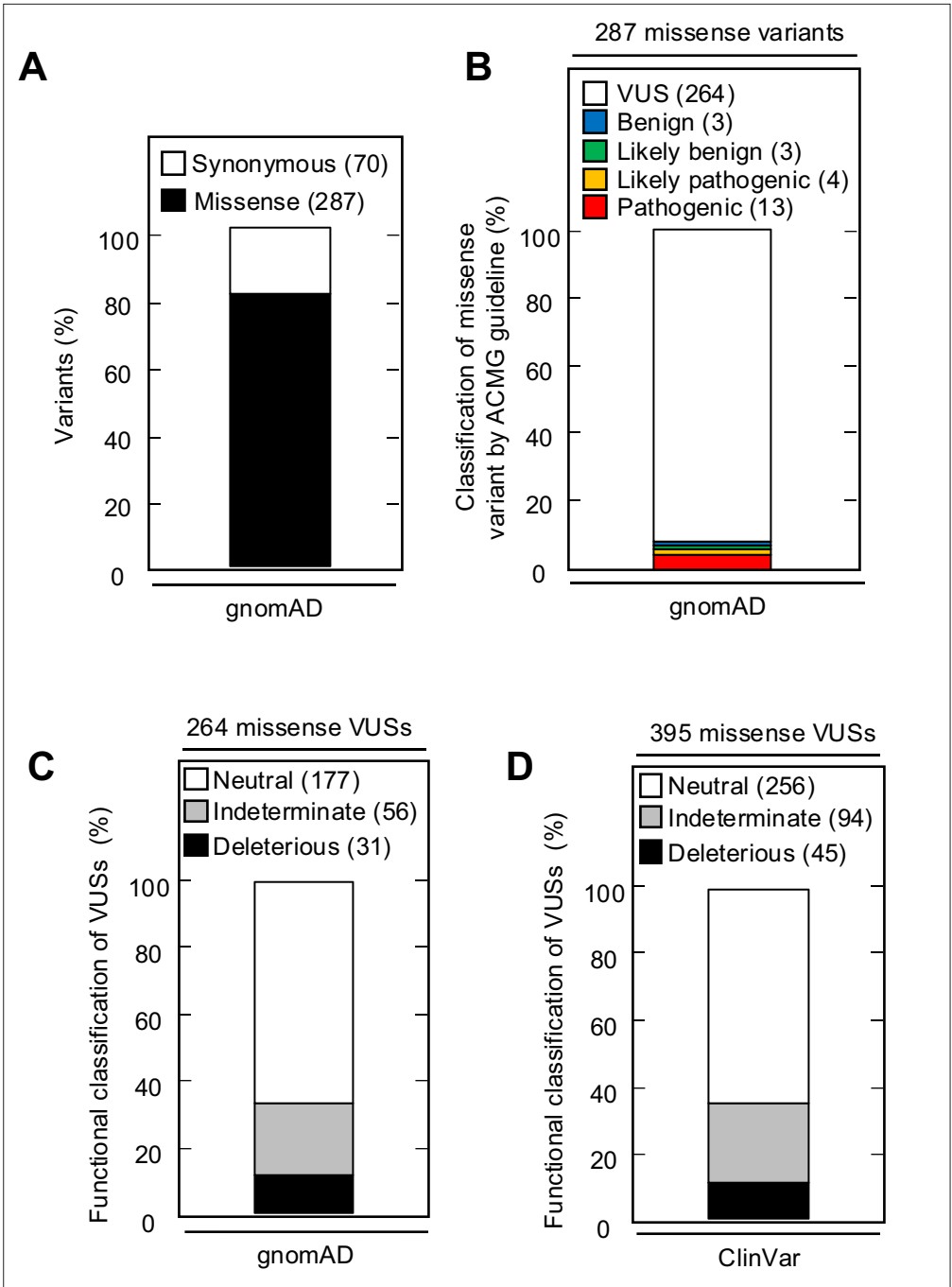

**Figure 5.** *CDKN2A* synonymous and missense variants reported in gnomAD and ClinVar. (**A**) Synonymous and missense variants in *CDKN2A* reported in gnomAD. (**B**) 287 *CDKN2A* missense variants reported in gnomAD, by American College of Medical Genetics (ACMG) guideline classification. (**C**) 264 missense variants in *CDKN2A* reported in gnomAD, by functional classification (deleterious – black box; indeterminate – gray box; neutral – white box). (**D**) 395 missense variants in *CDKN2A* reported in ClinVar, by functional classification (deleterious – black box; indeterminate – gray box; neutral – white box).

We characterized variants based upon a broad cellular phenotype, cell proliferation, in a single PDAC cell line. It is possible that *CDKN2A* variant functional classifications are cell-specific and assay-specific. Our assay may not encompass all cellular functions of CDKN2A and an alternative assay of a specific CDKN2A function, such as CDK4 binding, may result in different variant functional classifications. Furthermore, *CDKN2A* variants may have different effects if alternative cell lines are used

for the functional assay. However, cell-specific effects appear to be limited. In our previous study, we characterized 29 *CDKN2A* VUSs in three PDAC cell lines, using cell proliferation and cell cycle assays, and found agreement between all functional classifications (*Kimura et al., 2022*).

This study supports the utility of our in vitro functional assay. In general, we found that benchmark pathogenic variants, benchmark benign variants, and VUSs previously reported to be functionally deleterious had congruent functional classifications in our assay. Moreover, we found that functionally deleterious effects were enriched among somatic missense mutations, and depleted in missense VUSs in gnomAD, compared to all *CDKN2A* missense variants. Importantly, our functionally assay provides evidence to reclassify 301 of 395 (76.2%) missense VUSs reported in ClinVar and 208 of 264 (78.8%) missense VUSs reported in gnomAD. These include 45 (11.4%) VUSs in ClinVar and 31 missense VUSs in gnomAD that could be reclassified as likely pathogenic variants.

In this study, we determined functional classifications for all possible *CDKN2A* missense variants. Comparison of our functional classifications to in silico variant effect predictors, including recently described algorithms based on machine learning, provides performance benchmarks and supports current recommendations integrating data computational data into variant interpretation guidelines.

# Methods

**Key resources table**

| Reagent type (species) or resource | Designation | Source or reference | Identifiers | Additional information |
|---|---|---|---|---|
| Gene (*Homo sapiens*) | CDKN2A | GenBank | Gene ID: 1029 NM_000077.5 NP_000068.1 | |
| Cell line (*H. sapiens*) | PANC-1 | American Type Culture Collection | Cat. #: CRL-1469 RRID:CVCL_0480 | |
| Cell line (*H. sapiens*) | 293T | American Type Culture Collection | Cat. #: CRL-3216 RRID:CVCL_0063 | |
| Recombinant DNA reagent | pHAGE-CDKN2A (plasmid) | Addgene | RRID:Addgene_116726 | Lentiviral vector expressing CDKN2A |
| Recombinant DNA reagent | pLentiV_Blast (plasmid) | Addgene | RRID:Addgene_111887 | |
| Recombinant DNA reagent | pLJM1-Empty (plasmid) | Addgene | RRID:Addgene_91980 | |
| Recombinant DNA reagent | psPAX2 (plasmid) | Addgene | RRID:Addgene_12260 | |
| Recombinant DNA reagent | pCMV-VSV-G (plasmid) | Addgene | RRID:Addgene_8454 | |
| Recombinant DNA reagent | pLJM1-CDKN2A (plasmid) | Twist Bioscience | | Lentiviral vector expressing codon-optimized CDKN2A |
| Recombinant DNA reagent | pLJM1-CDKN2A-Leu32Leu (plasmid) | This paper | | Lentiviral vector expressing CDKN2A-Leu32Leu |
| Recombinant DNA reagent | pLJM1-CDKN2A-Leu32Pro (plasmid) | This paper | | Lentiviral vector expressing CDKN2A-Leu32Pro |
| Recombinant DNA reagent | pLJM1-CDKN2A- Gly101Gly (plasmid) | This paper | | Lentiviral vector expressing CDKN2A-Gly101Gly |
| Recombinant DNA reagent | pLJM1-CDKN2A- Gly101Trp (plasmid) | This paper | | Lentiviral vector expressing CDKN2A-Gly101Trp |
| Recombinant DNA reagent | pLJM1-CDKN2A- Val126Asp (plasmid) | This paper | | Lentiviral vector expressing CDKN2A-Val126Asp |
| Recombinant DNA reagent | pLJM1-CDKN2A- Val126Val (plasmid) | This paper | | Lentiviral vector expressing CDKN2A-Val126Val |

*Continued on next page*

*Continued*

| Reagent type (species) or resource | Designation | Source or reference | Identifiers | Additional information |
|---|---|---|---|---|
| Recombinant DNA reagent | pLentiV-Blast-CellTag (plasmid) | This paper | | Lentiviral vector expressing CellTAg |
| Sequence-based reagent | PCR primers | This paper | | See *Supplementary file 7* |
| Commercial assay or kit | GenePrint 10 System | Promega Corporation | Cat. #: B9510 | |
| Commercial assay or kit | PCR-based MycoDtect kit | Greiner Bio-One | Cat. #: 463 060 | |
| Commercial assay or kit | Q5 Site-Directed Mutagenesis kit | New England Biolabs | Cat. #: E0552 | |
| Commercial assay or kit | PureLink Genomic DNA Mini Kit | Invitrogen | Cat. #: K1820-01 | |
| Commercial assay or kit | KAPA HiFi HotStart PCR Kit | Kapa Biosystems | Cat. #: KK2501 | |
| Commercial assay or kit | Qubit dsDNA HS assay kit | Invitrogen | Cat. #: Q33230 | |
| Commercial assay or kit | MiSeq Reagent Kit v2 (300 cycles) | Illumina | Cat. #: MS-102-2002 | |
| Chemical compound, drug | Dulbecco's modified Eagle's medium | Gibco/Thermo Fisher | Cat. #: 11995-065 | |
| Chemical compound, drug | Lipofectamine 3000 Transfection Reagent | Thermo Fisher Scientific | Cat. #: L3000008 | |
| Chemical compound, drug | Q5 Hot Start High-Fidelity 2X Master Mix | New England Biolabs | Cat. #: M0494S | |
| Chemical compound, drug | Agencourt AMPure XP system | Beckman Coulter, Inc | Cat. #: A63881 | |
| Chemical compound, drug | Lenti-X Concentrator | Clontech | Cat. #: 631231 | |
| Chemical compound, drug | FxCycle Violet Ready Flow Reagent | Invitrogen | Cat. #: R37166 | |
| Software, algorithm | TC20 Automated Cell Counter | Bio-Rad Laboratories | Cat. #: 1450102 | |
| Software, algorithm | MiSeq System | Illumina | | |
| Software, algorithm | MiSeq control software | Illumina | | Version 2.5.0.5 |
| Software, algorithm | R | The R Foundation | RRID:SCR_001905 | Version 4.2.0 |
| Software, algorithm | JMP | SAS | RRID:SCR_014242 | Version 11 |
| Software, algorithm | Python statsmodel package | The Python Software Foundation | RRID:SCR_016074 | Version 0.14.0 |

### Cell lines

PANC-1 (American Type Culture Collection, Manassas, VA, USA; catalog no. CRL-1469), a human PDAC cell line with a homozygous deletion of *CDKN2A* (*Caldas et al., 1994*) and 293T (American Type Culture Collection; catalog no. CRL-3216), a human embryonic kidney cell line, were maintained in Dulbecco's modified Eagle's medium (Thermo Fisher Scientific Inc, Waltham, MA, USA; catalog no. 11995-065) supplemented with 10% fetal bovine serum (Thermo Fisher Scientific Inc; catalog no. 26140-079). Cell line authentication and mycoplasma testing were performed using the GenePrint 10 System (Promega Corporation, Madison, WI, USA; catalog no. B9510) and the PCR-based MycoDtect kit (Greiner Bio-One, Monroe, NC, USA; catalog no. 463 060) (Genetics Resource Core Facility, The Johns Hopkins University, Baltimore, MD, USA).

### CDKN2A somatic mutation data

CDKN2A (p16$^{INK4}$; NP_000068.1) missense somatic mutation data was obtained from the Catalogue Of Somatic Mutations In Cancer (*Forbes et al., 2010*), The Cancer Genome Atlas (*Muddabhaktuni and Koyyala, 2021*), patients with cancer undergoing sequencing at The Johns Hopkins University School of Medicine (Baltimore, MD, USA), Memorial Sloan Kettering-Integrated Mutation Profiling of Actionable Cancer Targets Clinical Sequencing Cohort (*Cheng et al., 2015*). CDKN2A variant data was obtained from gnomAD v.4.1.0 and ClinVar (*Landrum et al., 2014*).

### Plasmids

pHAGE-CDKN2A (Addgene, Watertown, MA, USA; plasmid no. 116726) was created by Gordon Mills & Kenneth Scott (*Ng et al., 2018*). pLJM1 (Addgene; plasmid no. 91980) was created by Joshua Mendell (*Golden et al., 2017*). pLentiV_Blast (Addgene, plasmid no. 111887) was created by Christopher Vakoc (*Tarumoto et al., 2020*). psPAX2 (Addgene, plasmid no. 12260) was created by Didier Trono, and pCMV-VSV-G (Addgene, plasmid no. 8454) was created by Bob Weinberg (*Stewart et al., 2003*).

### CDKN2A expression plasmid libraries

Codon-optimized CDKN2A cDNA using p16$^{INK4A}$ amino acid sequence (NP_000068.1) was designed (*Supplementary file 12*) and pLJM1 containing codon-optimized CDKN2A (pLJM1-CDKN2A) generated by Twist Bioscience (South San Francisco, CA, USA). 156 plasmid libraries were then synthesized by using pLJM1-CDKN2A, such that each library contained all possible 20 amino acids variants (19 missense and 1 synonymous) at a given position, generating 500 ng of each plasmid library (Twist Bioscience, South San Francisco, CA, USA). The proportion of variant in each library was shown in *Supplementary file 2*. Variants with a representation of less than 1% in a plasmid library were individually generated using the Q5 Site-Directed Mutagenesis kit (New England Biolabs, Ipswich, MA, USA; catalog no. E0552), and added to each library to a calculated proportion of 5%. Primers used for site-directed mutagenesis are given in *Supplementary file 13*. Each library was then amplified to generate at least 5 μg of plasmid DNA using QIAGEN Plasmid Midi Kit (QIAGEN, Germantown, MD, USA; catalog no. 12143).

### Single variant CDKN2A expression plasmids

Individual pLJM1-CDKN2A expression constructs for CDKN2A missense variants, p.L32L, p.L32P, p.G101G, p.G101W, p.V126D, and p.V126V, were generated using the Q5 Site-Directed Mutagenesis kit (New England Biolabs, Ipswich, MA, USA; catalog no. E0552). Primers used for site-directed mutagenesis are given in *Supplementary file 13*. Integration of each CDKN2A variant was confirmed using Sanger sequencing (Genewiz, Plainsfield, NJ, USA) using the CMV Forward sequencing primer (CGCA AATGGGCGGTAGGCGTG). The manufacturer's protocol was followed unless otherwise specified.

### CellTag plasmid library

Twenty nonfunctional 9 base pair barcodes 'CellTags' were subcloned into pLentiV_Blast using the Q5 Site-Directed Mutagenesis kit (New England Biolabs, Ipswich, MA, USA; catalog no. E0552) (*Biddy et al., 2018*). Primers used to generate each CellTag plasmid are given in *Supplementary file 13*. Integration of each CellTag was confirmed using Sanger sequencing (Genewiz) (sequencing primer: AACTGGGAAAGTGATGTCGTG). The manufacturer's protocol was followed unless otherwise specified. CellTag plasmids were then pooled to form a CellTag plasmid library with equal representation of each CellTag plasmid.

### Lentivirus production

Lentivirus production was performed as previously described with the following modifications (*Kimura et al., 2022*). pLJM1 lentiviral expression vectors (plasmid libraries and single variant expression plasmids) and lentiviral packaging vectors (psPAX2 and pCMV-VSV-G) were transfected into 293T cells using Lipofectamine 3000 Transfection Reagent (Thermo Fisher Scientific, Waltham, MA, USA; catalog no. L3000008). Media was collected at 24 hr and 48 hr, pooled, and lentiviral particles concentrated using Lenti-X Concentrator (Clontech, Mountain View, CA, USA; catalog no. 631231) using the manufacturer's protocol.

## Lentiviral transduction

PANC-1 cells were used for CDKN2A plasmid library and single variant CDKN2A expression plasmid transductions. PANC-1 cells previously transduced with pLJM1-CDKN2A (PANC-1$^{CDKN2A}$) and selected with puromycin were used for CellTag library transductions. Briefly, $1 \times 10^5$ cells were cultured in media supplemented with 10 µg/mL polybrene and transduced with $4 \times 10^7$ transducing units per mL of lenti-virus particles. Cells were then centrifuged at $1200 \times g$ for 1 hr. After 48 hr of culture at 37°C and 5% $CO_2$, transduced cells were selected using 3 µg/mL puromycin (CDKN2A plasmid libraries and single variant CDKN2A expression plasmids) or 5 µg/mL blasticidin (CellTag plasmid library) for 7 days. Expected MOI was one. After selection, cells were trypsinized and $5 \times 10^5$ cells were seeded into T150 flasks. DNA was collected from remaining cells and this sample was named as day 9. T150 flasks were cultured until confluent and then DNA was collected. The time for cells to become confluent varied for each amino acid residue (days 16–40, *Supplementary file 5*). DNA was extracted from PANC-1 cells using the PureLink Genomic DNA Mini Kit (Invitrogen, Carlsbad, CA, USA; catalog no. K1820-01). The assay for CellTag library was repeated in triplicate. We repeated our CDKN2A assay in duplicate for 28 residues. For the remaining 128 CDKN2A residues the assay was completed once.

## Generation of sequence libraries

Library preparation and sequencing was performed as previously described with the following modifications (*Kinde et al., 2011*). For the first stage PCR, three target specific primers were designed to amplify *CDKN2A* amino acid positions 1–53, 54–110, and 111–156 (*Supplementary file 13*). Forward and reverse first stage primers contained 5′ M13F (GTAAAACGACGGCCAGC) and M13R (CAGG AAACAGCTATGAC) sequence, respectively, to enable amplification and ligation of Illumina adapter sequences in a second stage PCR (*Supplementary file 13*). DNA was amplified with Q5 Hot Start High-Fidelity 2X Master Mix (New England Biolabs; catalog no. M0494S). For the first stage PCR, each DNA sample was amplified in three reactions each containing 66 ng of DNA for 18 cycles. First stage PCR products for each sample were then pooled and purified using the Agencourt AMPure XP system (Beckman Coulter, Inc, Brea, CA, USA; catalog no. A63881), eluting into 50 µL of elution buffer. Puri-fied PCR product was amplified in a second stage PCR to add Illumina adaptor sequences and indexes (*Supplementary file 13*). Second stage PCR Amplification was performed with KAPA HiFi HotStart PCR Kit (Kapa Biosystems, Wilmington, MA, USA; catalog no. KK2501) in 25 µL reactions containing 5X KAPA HiFi Buffer – 5 µL, 10 mM KAPA dNTP Mix – 0.75 µL, 10 µM forward primer – 0.75 µL, 10 µM reverse primer – 0.75 µL. For the first stage PCR, 66 ng of template DNA and 12.5 µL, Q5 Hot Start High-Fidelity 2X Master Mix was used with the following cycling conditions: 98°C for 30 s; 18 cycles of 98°C for 10 s, 72°C for 30 s, 72°C for 25 s; 72°C for 2 min. For the second stage PCR, 0.25 µL of first stage PCR product and 0.5 µL of 1 U/µL KAPA HiFi HotStart DNA Polymerase was used with the following cycling conditions: 95°C for 3 min; 25 cycles of 98°C for 20 s, 62°C for 15 s, 72°C for 1 min. Second stage PCR products were purified with the Agencourt AMPure XP system (Beckman Coulter, Inc; catalog no. A63881) into 30 µL of elution buffer. Samples were quantified by Qubit using dsDNA HS assay kit (Invitrogen; catalog no. Q33230).

## Sequencing and analysis

Sequence libraries were pooled in equimolar amounts into groups of 16 samples and sequenced on the Illumina MiSeq System (Illumina, San Diego, CA, USA) with the MiSeq Reagent Kit v2 (300 cycles) (Illumina catalog no. MS-102-2002) to generate 150 base pair paired-end reads. Samples were demultiplexed and FASTQ sequence read files were generated with MiSeq control software 2.5.0.5 (Illumina). Paired sequence reads were then combined into a single contiguous sequence using Paired-End Read Merger (*Zhang et al., 2014*). Reads supporting each variant at a given amino acid position were counted using Perl.

## Functional characterization of *CDKN2A* variants using a gamma GLM

We determined if a variant has a fitness advantage by assessing the significance of the observed ratio $r_{v,cf}$ at confluence between the number of cells with a missense variant $v$ and the number of cells with a synonymous variant at a given amino acid position. Using the missense variant as a benchmark variant, we assumed that the distribution of $r_{v,cf}$ can be explained by two key covariates: $r_{v,init}$, which represent the missense variant-to-synonymous variant ratio at day 9, and $p_{v,init}$, the proportion of the

missense variant cells among other variants, including the synonymous variant, at the studied position. More specifically, given the variables $r_{v,init} \wedge p_{v,init}$, the ratio at confluence follows a distribution:

$$r_{v,cf} \sim \Gamma\left(\alpha, \beta_v\right)$$

where the mean $u_v$ of the Gamma distribution is such that:

$$u_v = \frac{\alpha}{\beta_v} = r_{v,init}^a p_{v,init}^b.$$

Here, the parameters of the null model to estimate are $\alpha, a, and\, b$, where $\alpha$ is the shape parameter of the Gamma distribution and is assumed to be the same for all variants. This model is a gamma GLM over the response variable $r_{v,cf}$ with a log-link function and covariates $log\left(r_{v,init}\right)$ and $log\left(p_{v,init}\right)$. Estimating the parameters will provide a null distribution of $r_{v,cf}$, generating a p-value for every observed $r_{v,cf}$ for any variant at a given position.

To estimate the parameters $\alpha, a$, and $b$, we utilized three control experiments where the CellTag plasmid library was transduced into PANC-1[CDKN2Aco] cells so that each CellTag represented a neutral variant. For a single experiment, every variant can be considered as wild-type, and we test the other 19 variants against it, knowing that they are neutral and therefore follow the null distribution. This provides us with 19×20 triplets $\left(r_{v,cf}, p_v^{init}, r_v^{init}\right)$, for every experiment, yielding 1140 data points when considering all three experiments together. To estimate the parameters using these 1140 data points, we fit the GLM corresponding GLM model using the sklearn.linear_model module.

After the estimation of parameters $\alpha, a$, and $b$, every observation for a tested variant $v$ at a given position of the triplet $\left(r_{v,cf}, p_v^{init}, r_v^{init}\right)$ yields a p-value, defined as the probability of observing a ratio at confluence that is at least $r_{v,cf}$ given $p_{v,init}, r_{v,init}$ under the null Gamma model. As some variants were tested in repeated experiments, we combined their associated p-values into a single p-value using Fisher's method. Finally, to determine if a variant presents a fitness advantage, we apply a Benjamini-Hochberg estimator on all the tested variants p-values, fixing the false discovery rate at a level of 0.05.

## Functional characterization of CDKN2A variants using log$_2$ normalized fold change

Fold change for each variant was calculated using the proportion of total reads representing a variant at confluency (days 16–40, *Supplementary file 5*) to the proportion of total reads representing a variant on day 9 after transfection. Fold change was then normalized to the synonymous variant at each residue and then log$_2$ normalized fold change values calculated (*Supplementary file 4*, *Supplementary file 6*). Variants with log$_2$ normalized fold change values greater than or equal to the minimum value of benchmark pathogenic variants were characterized as functionally deleterious, while variants with values smaller than or equal to the maximum value of benchmark benign variants were characterized as functionally neutral (*Supplementary file 6*). Log$_2$ normalized fold change values between these defined thresholds were classified as indeterminate. Mean values were used for replicated variants.

## Data visualization

Heatmap of individual variant p-values by amino position was generated using R with the heatmaply package (*Galili et al., 2018*).

## Cell proliferation assay

Cell proliferation assay was performed as previously described with the following modifications (*Kimura et al., 2022*). $1 \times 10^5$ cells were seeded into in vitro culture (day 0). Cells were counted on day 14 using a TC20 Automated Cell Counter (Bio-Rad Laboratories, Hercules, CA, USA; catalog no. 1450102). Relative cell proliferation value was calculated as cell number normalized to empty vector control. Assays were repeated in triplicate. Mean cell proliferation value and standard deviation (s.d.) were calculated.

## Variant effect predictions

Publicly available algorithms were used to predict the consequence of *CDKN2A* missense variants. Prediction algorithms used included: CADD (*Kircher et al., 2014*), PolyPhen-2 (*Adzhubei et al.,*

*2010*), SIFT (*Kumar et al., 2009*), VEST (*Carter et al., 2013*), AlphaMissense (*Cheng et al., 2023*), ESM1b (*Brandes et al., 2023*), and PrimateAI-3D (*Gao et al., 2023*; *Supplementary file 7*). Poly-Phen-2, SIFT, VEST, AlphaMissense, and ESM1b prediction were available for all missense variants. CADD scores were available for 910 missense variants and where multiple CADD scores were possible, mean values were used. PrimateAI-3D prediction scores were available for 904 assayed missense variants.

## Statistical analyses

Statistical analyses were performed using JMP v.11 (SAS, Cary, NC, USA) and Python statsmodel package (version 0.14.0). Student's t-tests was used to compare mean cell proliferation values. A chi-square test was used to compare the proportion of functionally deleterious variants for variants present in <2% and ≥2% of the cell pool at day 9. A Fisher's exact test was used to compare prevalence of functionally deleterious *CDKN2A* variants in colorectal cancer cases from COSMIC with and without somatic mutations in mismatch repair genes. Z-tests with multiple test correction performed with the Bonferroni method was used in the following comparisons: (1) proportion of functionally deleterious variants present in <2% of the cell pool and ≥2% of the cell pool at day 9 binned in 1% intervals, (2) proportion of variants in each domain predicted to have deleterious or pathogenic effect by the majority of algorithms, (3) proportion of functionally deleterious variants in each domain, and (4) proportion of functionally deleterious missense variants and somatic mutations.

## Acknowledgements

National Institutes of Health grant P50CA62924 (NJR). Susan Wojcicki and Dennis Troper (NJR). The Sol Goldman Pancreatic Cancer Research Center (NJR). The Rolfe Pancreatic Cancer Foundation (NJR). The Japanese Society of Gastroenterology Support for Young Gastroenterologists Studying in the United States (HK). The Japan Society for the Promotion of Science Overseas Research Fellowships (HK).

## Additional information

### Competing interests

Cristian Tomasetti: Receives royalties from Exact Science; is a founder of C2T, a scientific advisor for PrognomiQ, and a consultant for Bayer. The other authors declare that no competing interests exist.

### Funding

| Funder | Grant reference number | Author |
| --- | --- | --- |
| National Cancer Institute | P50CA62924 | Nicholas Jason Roberts |
| Susan Wojcicki and Dennis Troper | | Nicholas Jason Roberts |
| The Sol Goldman Pancreatic Cancer Research Center | | Nicholas Jason Roberts |
| Rolfe Pancreatic Cancer Foundation | | Nicholas Jason Roberts |
| Japanese Society of Gastroenterology | Support for Young Gastroenterologists Studying in the United States | Hirokazu Kimura |
| Japan Society for the Promotion of Science | Overseas Research Fellowships | Hirokazu Kimura |

The funders had no role in study design, data collection and interpretation, or the decision to submit the work for publication.

## Author contributions
Hirokazu Kimura, Conceptualization, Data curation, Formal analysis, Investigation, Visualization, Methodology, Writing – original draft, Writing – review and editing; Kamel Lahouel, Conceptualization, Data curation, Formal analysis, Investigation, Methodology, Writing – review and editing; Cristian Tomasetti, Conceptualization, Resources, Data curation, Formal analysis, Investigation, Methodology, Writing – original draft, Writing – review and editing; Nicholas Jason Roberts, Conceptualization, Resources, Data curation, Formal analysis, Investigation, Methodology, Writing – original draft, Project administration, Writing – review and editing

## Author ORCIDs
Nicholas Jason Roberts  https://orcid.org/0000-0002-8709-0664

Reviewer #1 (Public review): https://doi.org/10.7554/eLife.95347.4.sa1
Author response https://doi.org/10.7554/eLife.95347.4.sa2

## Additional files

### Supplementary files
Supplementary file 1. Assay outputs for CellTag experiments.

Supplementary file 2. Proportion of each variant in the initial plasmid library.

Supplementary file 3. Proportion of each variant in residues R24, H66, and A127.

Supplementary file 4. Assay outputs and functional classifications for all possible *CDKN2A* missense and synonymous variants.

Supplementary file 5. Day of confluency by experiment and residue.

Supplementary file 6. Normalized fold change for all possible *CDKN2A* missense and synonymous variants.

Supplementary file 7. In silico variant effect predictions for *CDKN2A* missense variants.

Supplementary file 8. Assessment of in silico variant effect prediction models.

Supplementary file 9. Missense somatic mutations in *CDKN2A* reported in COSMIC, TCGA, JHU, MSK-IMPACT.

Supplementary file 10. *CDKN2A* missense and synonymous variants reported in gnomAD.

Supplementary file 11. *CDKN2A* missense variants of uncertain significance (VUSs) reported in ClinVar.

Supplementary file 12. Codon-optimized *CDKN2A* sequence.

Supplementary file 13. Sequences of primers used in study.

MDAR checklist

### Data availability
All data generated or analysed during this study are included in the manuscript and supporting files; source data files have been provided.

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
