## [Editor Report · eLife Assessment]

This is a saturation mutagenesis screening of CDKN2A gene, successfully assessing the functionality of the missense variants. The work is **solid** and well-prosecuted. The manuscript was improved during the revision process and this work will serve as a **valuable** resource for diagnostic labs as well as cancer geneticists.

---

## [Referee Report · Reviewer #1 (Public review)]

Summary:

Kimura et al performed a saturation mutagenesis study of CDKN2A to assess functionality of all possible missense variants and compare them to previously identified pathogenic variants. They also compared their assay result with those from in silico predictors.

Strengths:

CDKN2A is an important gene that modulate cell cycle and apoptosis, therefore it is critical to accurately assess functionality of missense variants. Overall, the paper reads well and touches upon major discoveries in a logical manner.

Weaknesses:

The paper lacks proper details for experiments and basic data, leaving the results less convincing. Analyses are superficial and does not provide variant-level resolution.

Comments on revisions

The manuscript was improved during the revision process.

---

## [Author Response]

The following is the authors’ response to the previous reviews

**Public Reviews:**

**Reviewer #1:**
Summary:Kimura et al performed a saturation mutagenesis study of CDKN2A to assess functionality of all possible missense variants and compare them to previously identified pathogenic variants. They also compared their assay result with those from in silico predictors.Strengths:CDKN2A is an important gene that modulate cell cycle and apoptosis; therefore it is critical to accurately assess functionality of missense variants. Overall, the paper reads well and touches upon major discoveries in a logical manner.Weaknesses:The paper lacks proper details for experiments and basic data, leaving the results less convincing. Analyses are superficial and does not provide variant-level resolution. Many of which were addressed during the revision process.Comments on revisions:The manuscript was improved during the revision process.

We thank the reviewer for their comments. We are grateful for the opportunity to provide additional information and data to clarify our approach and study results.

**Reviewer #2:**
Summary:This study describes a deep mutational scan across CDKN2A using suppression of cell proliferation in pancreatic adenocarcinoma cells as a readout for CDKN2A function. The results are also compared to in silico variant predictors currently utilized by the current diagnostic frameworks to gauge these predictors' performance. The authors also functionally classify CDKN2A somatic mutations in cancers across different tissues.Review:The goal of this paper was to perform functional classification of missense mutations in CDKN2A in order to generate a resource to aid in clinical interpretation of CDKN2A genetic variants identified in clinical sequencing. In our initial review, we concluded that this paper was difficult to review because there was a lack of primary data and experimental detail. The authors have significantly improved the clarity, methodological detail and data exposition in this revision, facilitating a fuller scientific review. Based on the data provided we do not think the functional characterization of CDKN2A variants is robust or complete enough to meet the stated goal of aiding clinical variant interpretation. We think the underlying assay could be used for this purpose but different experimental design choices and more replication would be required for these data to be useful. Alternatively, the authors could also focus on novel CDKN2A variants as there seems to be potential gain of function mutations that are simply lumped into "neutral" that may have important biological implications.Major concerns:Low experimental concordance. The p-value scatter plot (Figure 2 Figure Supplement 3A) across 560 variants shows low collinearity indicating poor replicability. These data should be shown in log2fold changes, but even after model fitting with the gamma GLM still show low concordance which casts strong doubt on the function scores.

Concordance among non-significant p-values is generally low because most of the signal comes from random variability across repeats. If the observed log2 fold change between the repeats is entirely due to noise, one would expect two repeated p-values to behave like independent random uniforms. True concordance is typically more evident in significant p-values because they reflect consistent effects above random noise. Functionally deleterious variants are called when their associated p-value is significant. To confirm this statement, a scatter plot with the log2 normalized fold change was added in Figure 2 Supplement 3C. We see low concordance between repeats in the log2 normalized fold changes centered around 0, corresponding to log log2 normalized changes mainly due to noise. The concordance increases as the variants become significant. One can notice that the correlation coefficient between duplicate assay results was almost identical between the model-based p-values and log2normalized fold change (Figure 2-figure supplement 3A and 3C, Appendix 1-table 4, and Appendix 1-table 6). Also, importantly, no variant was functionally deleterious in one replicate and functionally neutral in another, implying a perfect concordance in calls if we exclude variants that were called indeterminate in one of the two repeats. Finally, of variants with discordant classifications, only 6/560 repeats (1.1%) were functionally deleterious (significant p-value) in one replicate and of indeterminate function in another. We have updated the text as follows:

“Of variants with discordant classifications, 6 (1.1%) were functionally deleterious in one replicate and of indeterminate function in another. While 102 variants (18.2%) were functionally neutral in one replicate and of indeterminate function in another. Importantly, no variant that was functionally deleterious in one replicate and functionally neutral in another (Appendix 1 -table 4). Furthermore, the correlation coefficient between duplicate assay results was similar using the gamma GLM and log2 normalized fold change (Figure 2-figure supplement 3A and 3C).”

The more detailed methods provided indicate that the growth suppression experiment is done in 156 pools with each pool consisting of the 20 variants corresponding to one of the 156 aa positions in CKDN2A. There are several serious problems with this design.Batch effects in each of the pools preventing comparison across different residues. We think this is a serious design flaw and not standard for how these deep mutational scans are done. The standard would be to combine all 156 pools in a single experiment. Given the sequencing strategy of dividing up CDKN2A into 3 segments, the 156 pools could easily have been collapsed into 3 (1 to 53, 54 to 110, 111 to 156). This would significantly minimize variation in handling between variants at each residue and would be more manageable for performance of further replicates of the screen for reproducibility purposes. The huge variation in confluency time 16-40 days for each pool suggest that this batch effect is a strong source of variation in the experiment.

While there is variation in time to confluency between different amino acid residues, we do not anticipate this batch effect to significantly affect variant classifications in our study. For example, our results were generally consistent with previous classifications. All synonymous variants (one per residue) and benchmark benign variants assayed were classified as functionally neutral. Furthermore, of benchmark pathogenic variants assayed, none were classified as functionally neutral. 84% were classified as functionally deleterious and 16 percent were classified as indeterminate function.

Lack of experimental/biological replication: The functional assay was only performed once on all 156 CDKN2A residues and was repeated for only 28 out of 156 residues, with only ~80% concordance in functional classification between the first and second screens. This is not sufficiently robust for variant interpretation. Why was the experiment not performed more than once for most aa sites?

In our study we determined functional classifications for all *CDKN2A* missense variants while assessing variability with replicates across 28 residues. Of these variants, only 6 (1.1%) were functionally deleterious in one replicate and of indeterminate function in another. Furthermore, no variant was functionally deleterious in one replicate and functionally neutral in another (Appendix 1 -table 4). As noted above, we provided additional context in the manuscript.

For the screen, the methods section states that PANC-1 cells were infected at MOI=1 while the standard is an MOI of 0.3-0.5 to minimize multiple variants integrating into a single cell. At an MOI = 1 under a Poisson process which captures viral integration, ~25% of cells would have more than 1 lentiviral integrant. So in 25% of the cells the effect of a variant would be confounded by one or more other variants adding noise to the assay.

As noted previously, we are not able to differentiate effects due to multiple viral integrations per cells. However, we do not anticipate multiple viral integrations to significantly affect variant classifications in our study as our results are consistent with previous classifications, as described above.

While the authors provide more explanation of the gamma GLM, we strongly advise that the heatmap and replicate correlations be shown with the log2 fold changes rather than the fit output of the p-values.

Thank you for the suggestion. As noted, we provide additional explanation in the manuscript about why we classified variants using a gamma GLM. Using a gamma GLM, classification thresholds were determined using the change in representation of 20 non-functional barcodes in a pool of PANC-1 cells stably expressing CDKN2A after a period of in vitro proliferation. Our variant classifications were therefore not based on assay outputs for previously reported – benchmark – pathogenic or begin variants to determine thresholds. We strongly prefer using p-values and classifications using the gamma GLM in the manuscript. However, comparison of assay outputs using a gamma GLM and log2 fold change are included in the manuscript. Read counts, log2 fold change, and classifications based on log2 fold change are presented in the manuscript, for all variants. Readers who wish to use these data may do so and we refer them to the manuscript text, Appendix 1 -table 4, Appendix 1 -table 6, and Figure 2 -figure supplement 2.

In this study, the authors only classify variants into the categories "neutral", "indeterminate", or "deleterious" but they do not address CDKN2A gain-of-function variants that may lead to decreased proliferation. For example, there is no discussion on variants at residue 104, whose proliferation values mostly consist of higher magnitude negative log2fold change values. These variants are defined as neutral but from the one replicate of the experiment performed, they appear to be potential gain-of-function variants.

We have added a comment to the discussion to highlight that we did not identify potential gain-of-function variants. Specifically:

“We classified *CDKN2A* missense variants using a gamma GLM, as either functionally deleterious, indeterminate functional or functionally neutral. However, we did not classify variants that may have gain-of-function effects, resulting in decreased representation in the cell pool. Future studies are necessary to determine the prevalence and significance of *CDKN2A* gain-of-function variants.”

Minor concerns:The differentiation between variants of "neutral" and "indeterminate" function seems unnecessary and it seems like there are too many variants that fall into the "indeterminate" category. The authors seem to have set numerical thresholds for CDKN2A function using benchmark variants of known function. While the benchmark variants are important as a frame of reference for the "dynamic range" of the assay, their function scores should not necessarily be used to define hard cutoffs of whether a variant's function score can be interpreted.

We did not utilize benchmark variants to define thresholds for functional classifications using a gamma GLM. This is one of the strengths of using a gamma GLM model for classification. As explained in our manuscript, classification thresholds were determined using the change in representation of 20 non-functional barcodes in a pool of PANC-1 cells stably expressing CDKN2A after a period of in vitro proliferation. Our variant classifications were therefore not based on assay outputs for previously reported – benchmark – pathogenic or begin variants. While not required when using a gamma GLM, we included indeterminate classifications, which are not uncommon.

Figure 2 supplement 2 - on the x-axis, should "intermediate" be "indeterminate"?

This, and a similar typographical error in Figure 2 -figure supplement 3, has been corrected.